# Collateral sensitivity associated with antibiotic resistance plasmids

**Cristina Herencias[1†], Jerónimo Rodríguez-Beltrán[1,2†*], Ricardo León-Sampedro[1,2], Aida Alonso-del Valle[1], Jana Palkovičová[3], Rafael Cantón[1,4], Álvaro San Millán[1,2,5*]**

[1]Servicio de Microbiología. Hospital Universitario Ramón y Cajal and Instituto Ramón y Cajal de Investigación Sanitaria, Madrid, Spain; [2]Centro de Investigación Biológica en Red Epidemiología y Salud Pública, Instituto de Salud Carlos III, Madrid, Spain; [3]Department of Biology and Wildlife Diseases, Faculty of Veterinary Hygiene and Ecology, University of Veterinary and Pharmaceutical Sciences, Brno, Czech Republic; [4]Red Española de Investigación en Patología Infecciosa. Instituto de Salud Carlos III, Madrid, Spain; [5]Centro Nacional de Biotecnología-CSIC, Madrid, Spain

**Abstract** Collateral sensitivity (CS) is a promising alternative approach to counteract the rising problem of antibiotic resistance (ABR). CS occurs when the acquisition of resistance to one antibiotic produces increased susceptibility to a second antibiotic. Recent studies have focused on CS strategies designed against ABR mediated by chromosomal mutations. However, one of the main drivers of ABR in clinically relevant bacteria is the horizontal transfer of ABR genes mediated by plasmids. Here, we report the first analysis of CS associated with the acquisition of complete ABR plasmids, including the clinically important carbapenem-resistance conjugative plasmid pOXA-48. In addition, we describe the conservation of CS in clinical *E. coli* isolates and its application to selectively kill plasmid-carrying bacteria. Our results provide new insights that establish the basis for developing CS-informed treatment strategies to combat plasmid-mediated ABR.

**\*For correspondence:**
jeronimo.rodriguez.beltran@
gmail.com (JR-B);
asanmillan@cnb.csic.es (ÁSM)

[†]These authors contributed
equally to this work

**Competing interests:** The
authors declare that no
competing interests exist.

**Reviewing editor:** Marc
Lipsitch, Harvard TH Chan
School of Public Health, United
States

## Introduction

The rapid evolution of antibiotic resistance (ABR) in bacteria reduces the utility of clinically relevant antibiotics, making ABR one of the major challenges facing public health (*Jim, 2016*; *MacLean and San Millan, 2019*). There is therefore an urgent need for new treatment strategies to fight against resistant pathogens (*Baym et al., 2016*). Among the most promising alternatives, much attention has focused on collateral sensitivity (CS). CS occurs when the acquisition of resistance to one antibiotic causes increased susceptibility to a second antibiotic (*Szybalski and Bryson, 1952*) and can be exploited for the design of multi-drug strategies that select against ABR (*Imamovic and Sommer, 2013*; *Lázár et al., 2013*). Moreover, recent evidence suggests that the development of resistance might be prevented by treatments based on the sequential cycling of antibiotics whose resistance mechanisms produce reciprocal CS (*Imamovic and Sommer, 2013*). Several studies have cataloged CS networks emerging from mutations in chromosomal (*Imamovic and Sommer, 2013*; *Lázár et al., 2013*; *Maltas and Wood, 2019*; *Podnecky et al., 2018*; *Roemhild et al., 2020*) and plasmid genes (Dortet et al., 2015; Fröhlich et al., 2019; *Rosenkilde et al., 2019*), but the CS effects produced by the horizontal acquisition of ABR plasmids remain poorly understood.

Mobile genetic elements, especially plasmids, play a crucial role in the dissemination of ABR genes between clinical pathogens and are one of the major drivers of the alarming worldwide rise in ABR (*Partridge et al., 2018*). Despite the evolutionary advantage conferred by plasmids in the presence of antibiotics, plasmid acquisition tends to produce common metabolic alterations in the host bacterium (*San Millan et al., 2018*), that typically translate into a fitness cost (*San Millan and*

*MacLean, 2017*). We reasoned that the physiological alterations produced upon acquisition of ABR plasmids might lead to exploitable CS responses. To test this hypothesis, we analyzed the collateral effects associated with the acquisition of six natural plasmids carrying clinically relevant ABR genes in *Escherichia coli* MG1655. Our results reveal that most ABR plasmids, including the clinically important carbapenem-resistance conjugative plasmid pOXA-48, produce CS events of moderate effect. To extend our results, we explored the degree of conservation of CS responses associated with pOXA-48 across phylogenetically diverse wild-type *E. coli.* Our results suggest that these responses can be exploited to preferentially kill plasmid-carrying bacteria.

## Results and discussion

### CS induced by ABR plasmids

To test whether ABR plasmids elicit exploitable CS to other antibiotics, we selected six clinically relevant ABR plasmids (see Materials and methods, *Table 1* and *Figure 1—figure supplement 1*). All plasmids carried important ß-lactamase genes, including extended-spectrum ß-lactamases (ESBLs), AmpC-type ß-lactamases, or carbapenemases, as well as other relevant ABR genes (*Table 1* and *Figure 1—figure supplement 1*). These plasmids belonged to a broad range of incompatibility groups, covered a range of different sizes, included conjugative and non-conjugative specimens, and produced variable fitness effects, ranging from 10% to 27% reductions in relative fitness, when introduced into *E. coli* MG1655 (*Figure 1A*).

As a preliminary screening, we performed dose-response experiments with 13 antibiotics from eight drug families. For this analysis, we determined the minimal inhibitory concentration (MIC, see Materials and methods) using the broth microdilution method for plasmid-free MG1655 and each of its six plasmid-carrying derivatives (*Supplementary file 1A*). We analyzed the collateral antibiotic susceptibility effects associated with the presence of each plasmid, measured as the fold-change in the antibiotic MIC between plasmid-carrying and plasmid-free bacteria. As expected from their genetic content, all plasmids conferred resistance to ß-lactam antibiotics and most of them conferred additional resistance to other antibiotics from unrelated families (*Figure 1B* and *Supplementary file 1A*). Initially, we defined CS events as those plasmid-antibiotic combinations showing a minimum twofold reduction in MIC (calculated as the fold-change of the median MIC value obtained from four to five independent MIC determinations). We then performed additional MIC determinations (up to 10 replicates each) for the antibiotic-plasmid combinations that fulfilled this criterion to increase the robustness of our analysis (*Supplementary file 1A*). This led to the identification of seven CS instances producing a consistent ≥2-fold reduction in antibiotic susceptibility, of which six showed significant reductions in antibiotic susceptibility (pOXA-48–azithromycin, pCF12–azithromycin, pOXA-48–colistin, pCF12–kanamycin, pKAZ3–kanamycin, and pKAZ3-gentamicin; Mann-Whitney U test; p<0.015 in all cases; *Figure 1B*). Of note, CS to tetracycline elicited by

**Table 1.** Plasmids used in this study.

| Plasmid | Incompatibility group | Mobility (MOB family) | Size (bp) | Resistance Genes | Reference (Genbank Ac. No.) |
|---|---|---|---|---|---|
| pOXA-48_K8 | IncL | Conjugative (MOBP) | 65,499 | $bla_{OXA-48}$ | (*León-Sampedro et al., 2020*) (MT441554) |
| pKAZ3 | IncA/C | Conjugative (MOBH) | 147,957 | $bla_{VEB-9}$, qnrVC-1, sul1, tetA', tetC, dfrA1, dfrA23 | (*Flach et al., 2015*) (KR827392.1) |
| pKA2Q | IncQ | Mobilizable (MOBF) | 8,789 | $bla_{FOX-8}$ | (*Hernández-García et al., 2018a*) (MT720904) |
| pCF12 | IncX3 | Conjugative (MOBP) | 43,704 | $bla_{SHV-12}$, qnrS1 | This work (MT720906) |
| pCEMR | ColE-like | Mobilizable | 14,478 | $bla_{VIM-1}$, aac(6')-Ib3 sul1, aadA1, qacEΔ1 | (*Papagiannitsis et al., 2018*) (MG049738.1) |
| pKP-M1144 | ColE-like | Non-transmissible | 12,029 | $bla_{BEL-1}$, $bla_{GES-5}$, $bla_{IMP-8}$, aac(6')-Ib3 | (*Papagiannitsis et al., 2015*) (KF745070.2) |

[*]pOXA-48_K8 is a pOXA-48-like plasmid (see Materials and methods and *Figure 1—figure supplement 1*). For simplicity, we refer to this plasmid as pOXA-48 throughout the study.

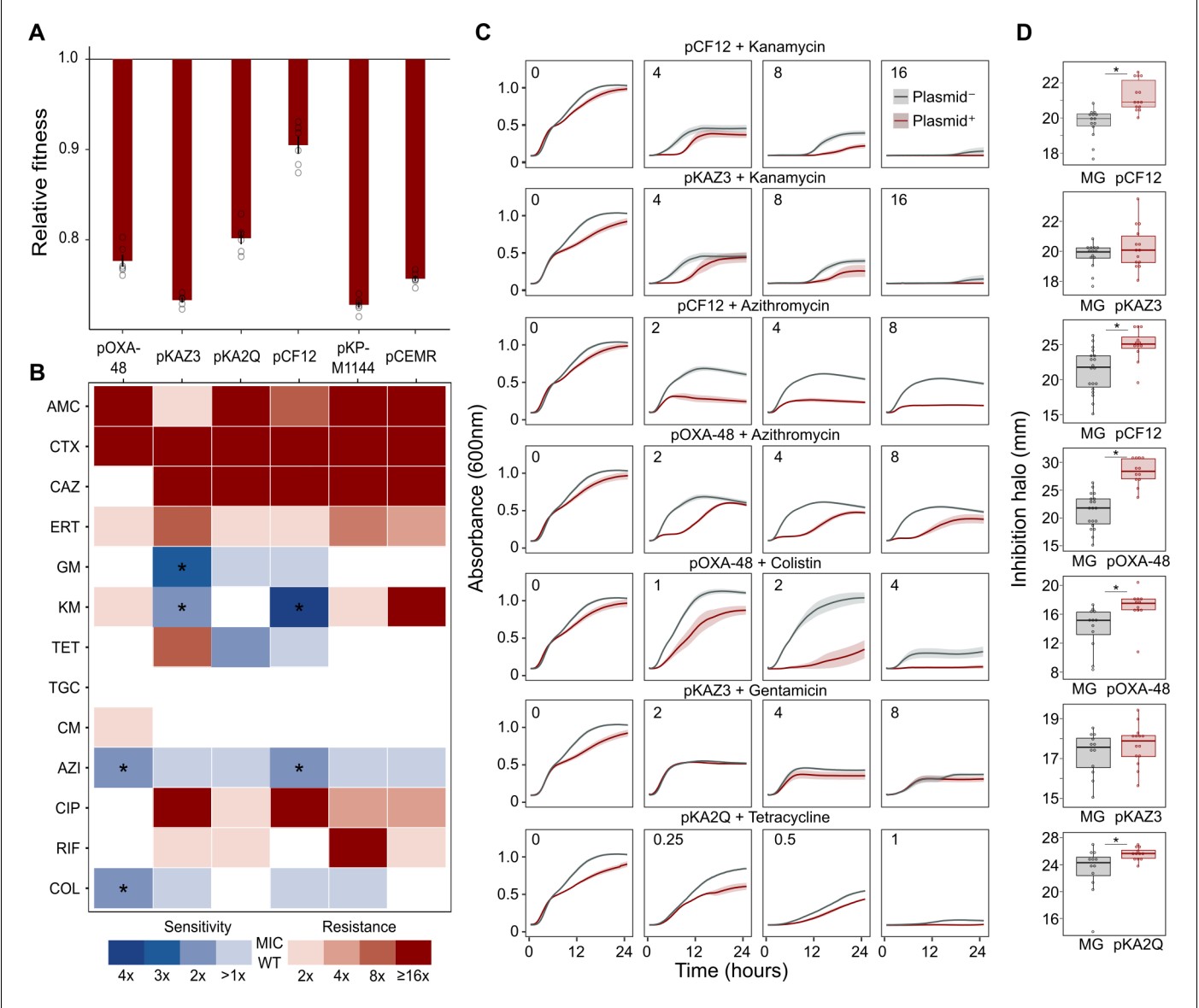

**Figure 1.** Collateral sensitivity and fitness effects associated with antibiotic resistance (ABR) plasmids. (A) Competition assay determined fitness of plasmid-carrying clones relative to the plasmid-free MG1655 strain. Bars represent the means of six independent experiments (represented by dots), and error bars represent the standard error of the mean. (B) Heat-map representing collateral responses to antibiotics associated with plasmid acquisition. The color code represents the fold change of MIC in plasmid-carrying derivatives compared to plasmid-free MG1655 (see legend). Asterisks indicate a significant ≥2 fold decrease in MIC (Mann-Whitney U test; p<0.015). AMC, amoxicillin-clavulanic acid; CTX, cefotaxime; CAZ, ceftazidime; ERT, ertapenem; GM, gentamicin; KM, kanamycin; TET, tetracycline; TGC, tigecycline; CM, chloramphenicol; AZI, azithromycin; CIP, ciprofloxacin; RIF, rifampicin; and COL, colistin. (C) Growth curves of MG1655 and plasmid-carrying MG1655 strains exposed to increasing antibiotic concentrations. The antibiotic concentration, in mg/L, is indicated at the top left corner of each panel. Lines represent the mean of 6 biological replicates, and the shaded area indicates standard error of the mean. (D) Boxplot representations of the inhibition halo diameters, in mm, obtained from disk-diffusion antibiograms of plasmid-free and plasmid-carrying MG1655. Plasmid/antibiotic combinations in each row are the same as the ones indicated in panel C. Horizontal lines within boxes indicate median values, upper and lower hinges correspond to the 25th and 75th percentiles, and whiskers extend to observations within 1.5 times the interquartile range. Individual data points are also represented (11–15 biological replicates). Asterisks indicate statistically significant differences (unpaired t-test with Welch's correction p<0.044).

The online version of this article includes the following source data and figure supplement(s) for figure 1:

**Source data 1.** Numerical data that are represented in *Figure 1*.

**Figure supplement 1.** Plasmids used in this study.

**Figure supplement 2—source data 1.** Numerical data that are represented in *Figure 1—figure supplement 1*.

**Figure supplement 2.** Analysis of bacterial growth curves.

pKA2Q did not reach statistical significance (Mann-Whitney U test; p=0.2), yet we decided to maintain it for subsequent analysis.

To further validate MIC results, we conducted two complementary series of experiments for the seven cases of plasmid-induced CS: (i) complete growth curves under a range of antibiotic concentrations and (ii) disk-diffusion antibiograms on agar plates. Analysis of the area under the growth curve, which integrates all growth parameters (maximum growth rate, lag duration and carrying capacity) (*DelaFuente et al., 2020*), confirmed significant plasmid-mediated CS effects at different antibiotic concentrations for six of the combinations (pOXA-48–azithromycin, pCF12–azithromycin, pOXA-48–colistin, pCF12–kanamycin, pKAZ3–kanamycin, and pKA2Q–tetracycline; *Figure 1C*, *Figure 1—figure supplement 2*, and *Supplementary file 1B*). The disk-diffusion results largely agreed with the MIC and growth-curve data, showing a general trend toward an increased diameter of the antibiotic inhibition zone in plasmid-carrying strains compared with plasmid-free MG1655 (*Figure 1D*). These differences were statistically significant for pCF12–kanamycin, pCF12–azithromycin, pOXA-48–azithromycin, pOXA-48–colistin, and pKA2Q–tetracycline (unpaired t-test with Welch's correction; p<0.044 in all cases). Overall, six of the seven CS instances initially identified were validated with at least one of the additional methods (growth curves or disk-diffusion assays), and four of them showed full agreement across all three methods. The disk-diffusion assays produced notably robust results with minimum hands-on time, suggesting that this technique is appropriate for screening CS responses in large strain collections. Together, these results demonstrate that ABR plasmids produce moderate but significant CS to different antibiotics, comparable to that generated by ABR chromosomal mutations in *E. coli* (*Imamovic and Sommer, 2013*; *Lázár et al., 2013*; *Podnecky et al., 2018*).

## Conservation of pOXA-48-mediated CS

The success and applicability of CS-informed therapeutic strategies are crucially dependent on the conservation of CS across diverse genetic contexts (*Podnecky et al., 2018*). To address the phylogenetic preservation of plasmid-induced CS, we focused on the plasmid pOXA-48, which confers resistance to last-resort carbapenem antibiotics, and whose prevalence in clinical settings is rising alarmingly (*David et al., 2019*). We tested the degree of conservation of pOXA-48-mediated CS patterns (to azithromycin and colistin) across phylogenetically diverse *E. coli* strains using disk-diffusion assays. We determined antibiotic susceptibility in nine diverse *E. coli* clinical isolates (*Figure 2A*) and their pOXA-48–carrying derivatives. CS to azithromycin showed a striking degree of conservation across the strains, with eight out of nine plasmid-carriers showing significantly higher susceptibility than their plasmid-free counterparts. In contrast, CS to colistin was conserved in only two of the tested strains (*Figure 2B*). Nevertheless, all strains showed CS responses to at least one antibiotic, suggesting that pOXA-48 elicits CS responses that could potentially be exploited therapeutically.

## Exploitation of CS to selectively kill pOXA-48-carrying *E. coli*

We next tested the potential of exploiting CS effects to kill plasmid-carrying populations. We rationally selected four *E. coli* isolates and their pOXA-48 carrying derivatives showing the following defined CS profiles: to colistin only (Ec10), to azithromycin only (Ec21 and Ec25), or to both antibiotics (Ec18). MG1655 was included as a control. Plasmid-bearers and plasmid-free strains were propagated separately in the presence of a single antibiotic or with sequential exposure to both antibiotics over 2 days. On the first day of the experiment, 24 bacterial populations of each strain were inoculated into media containing either colistin (4 mg/L) or azithromycin (8 mg/L). After 22 hr, populations able to grow were transferred to fresh media containing colistin or azithromycin in a full factorial experimental design. This approach gave rise to four antibiotic treatments: two single-drug treatments (denoted Azi→ Azi and Col→ Col) and two treatments in which the antibiotic changed (Azi→ Col and Col→ Azi). In 11 of the 20 possible strain-treatment combinations, pOXA-48–carrying populations showed higher mortality rates than their plasmid-free counterparts (*Figure 3*; log-rank test p<0.001 in all cases). As we used fixed antibiotic concentrations, the results of this experiment are likely to depend on the interplay between the particular resistance levels of each strain and the order in which the antibiotics are used. This may explain why we failed to observe significant differences in the cases in which the antibiotic treatment produced a sudden drop in viability for both plasmid-bearers and plasmid-free strains. Nevertheless, the mortality patterns observed were

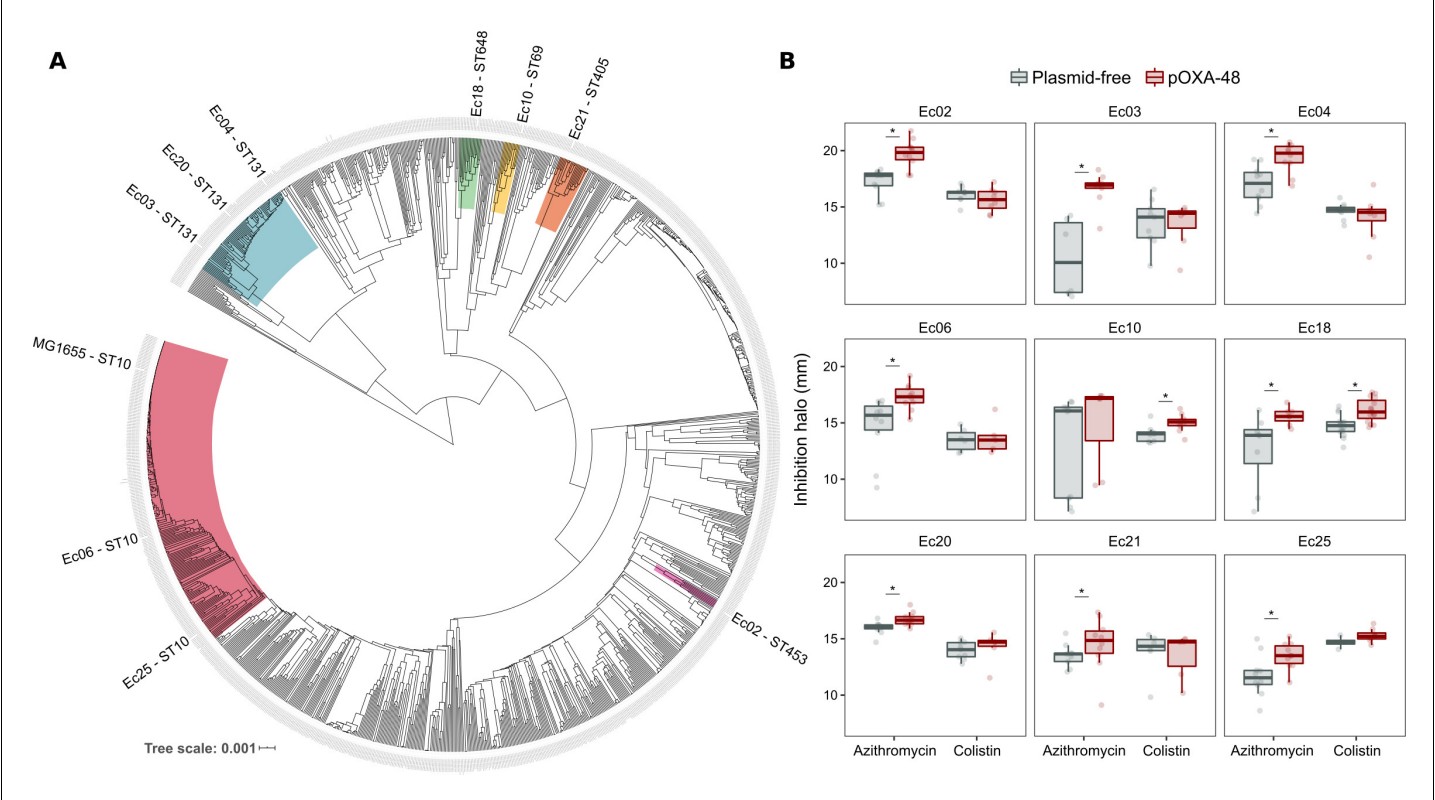

**Figure 2.** Phylogeny and collateral sensitivity profiles of *E. coli* clinical isolates. (A) Phylogenetic tree of *E. coli* species, highlighting the strains used in this study and their sequence types (ST). Colors within tree branches represent different STs. The tree depicts the phylogenetic relationships of 1344 representative *E. coli* genomes obtained from NCBI. (B) Representation of the inhibition halo diameters, in mm, obtained from disk-diffusion antibiograms of clinical isolates and their transconjugants carrying pOXA-48. Horizontal lines inside boxes indicate median values, upper and lower hinges correspond to the 25th and 75th percentiles, and whiskers extend to observations within 1.5 times the interquartile range. Individual data points are also represented (6–12 replicates, jittered to facilitate data visualization). Asterisks denote statistically significant differences (t-test with Welch's correction p<0.035 in all cases).

The online version of this article includes the following source data for figure 2:

**Source data 1.** Numerical data that are represented in *Figure 2*.

consistent with the CS patterns obtained in the disk-diffusion technique (*Figure 2B*). For instance, plasmid-carrying MG1655 and Ec18 strains, which show CS to both antibiotics, exhibited significant reductions in survival in all treatments. In contrast, for strain Ec10, which shows CS exclusively to colistin, selective killing was observed only when the treatment included colistin as the first antibiotic (Col→ Col or Col→ Azi).

## Concluding remarks

Our results reveal that the acquisition of clinically-relevant ABR plasmids induces CS to unrelated antibiotics. These findings thus serve as a stepping stone toward the development of new approaches aimed at blocking plasmid-mediated horizontal spread of ABR genes. These anti-plasmid strategies could help to tackle the alarming clinical and community spread of ABR. However, the molecular basis of plasmid-associated CS is currently unknown. The feasibility of rational broad-spectrum anti-plasmid strategies will depend on gaining comprehensive knowledge about the molecular mechanisms that increase antibiotic susceptibility upon plasmid acquisition. Until these therapies are available, CS-informed empirical treatment of plasmid-carrying bacteria has the potential to help resolve antibiotic resistant infections. In this regard, our results suggest that colistin and azithromycin may be valuable antibiotics for the treatment of pOXA-48–carrying enterobacteria. Both antibiotics are already used to treat enterobacterial infections (*Lübbert, 2016*; *Morrill et al., 2015*), and colistin in particular is currently used as a last resort antibiotic against carbapenem-

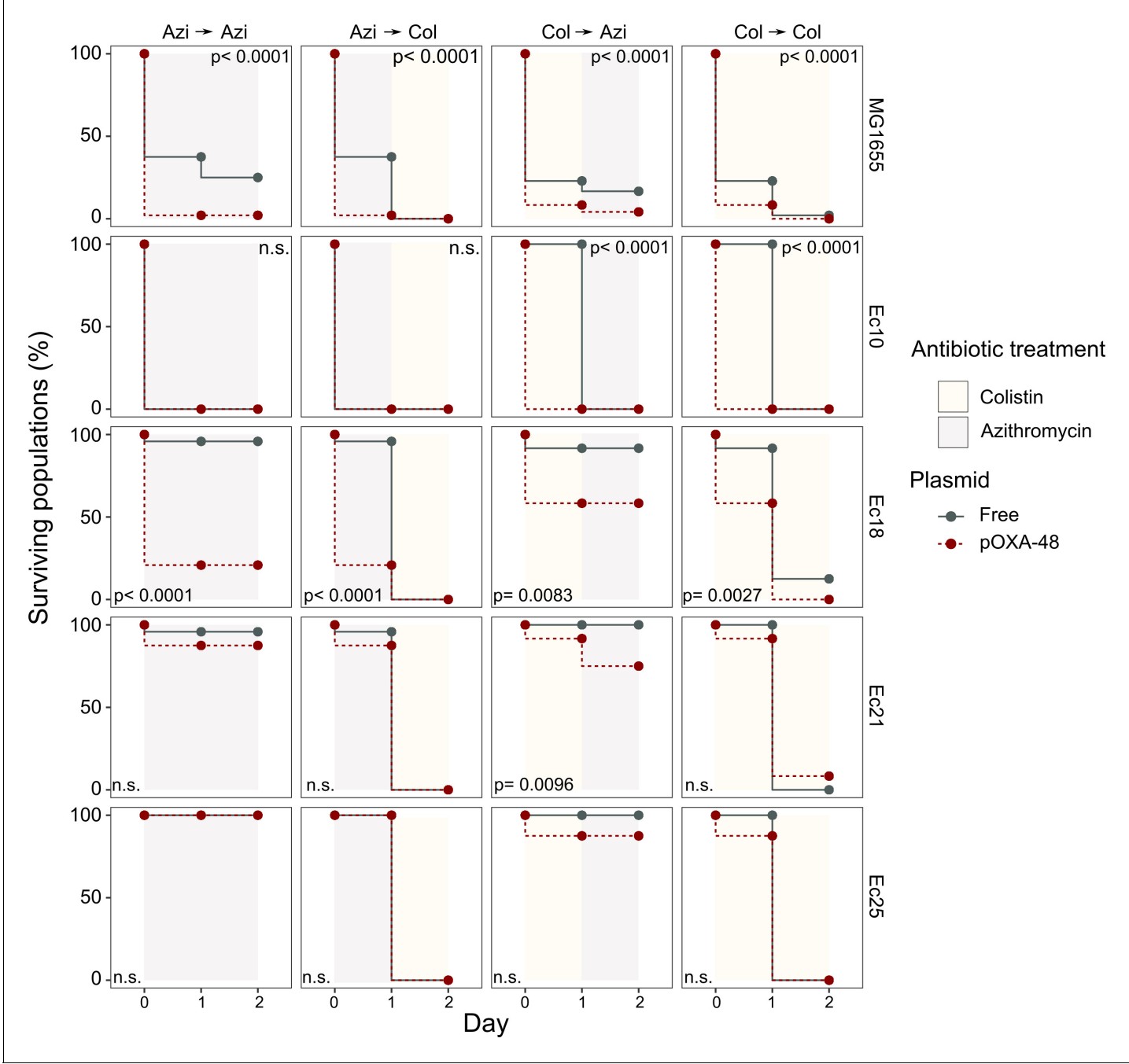

**Figure 3.** Plasmid-mediated CS can be exploited to preferentially kill plasmid-carrying bacteria. Survival curves of plasmid-free (gray solid lines) and plasmid-carrying (red dashed lines) bacterial populations in response to constant or sequential antibiotic treatments. Plasmid-free or plasmid-carrying bacterial populations were cultured separately for 2 days with either constant antibiotic exposure or with a change of antibiotic after the first day; treatments are specified by the background colors in each panel. Log-rank test p-values assessing statistically different survival curves are shown. n.s. non-significant. For strains Ec10, Ec18, Ec21, and Ec25 n = 24; for MG1665 n = 48. AZI, azithromycin; COL, colistin.

The online version of this article includes the following source data for figure 3:

**Source data 1.** Numerical data that are represented in *Figure 3*.

resistant enterobacteria (*Rodríguez-Baño et al., 2018*). A limitation of our experimental setup is that plasmid-bearers and plasmid-free populations were subjected to antibiotic treatments separately. More experiments are thus required to confirm that CS-informed treatments can eradicate plasmid-carrying bacteria from mixed populations and to gauge the clinical relevance of our results.

In addition, future studies should focus on determining the best treatment option. Sequential treatments offer a promising strategy, as they better exploit evolutionary trade-offs (such as CS) to constraint ABR development (*Kim et al., 2014*; *Roemhild and Schulenburg, 2019*). Combination therapies might also provide a valuable strategy as long as both antibiotics act synergistically to inhibit bacterial growth (*Tamma et al., 2012*). In summary, although extensive work will be needed before these therapies are available, the findings presented here open new avenues of research aimed at understanding plasmid-associated CS and pave the way for the development of new CS-based treatment strategies to combat plasmid-mediated ABR.

## Materials and methods

### Bacterial strains, plasmids, and culture conditions

All experiments were performed in liquid Lennox lysogeny broth (LB; CONDA) or on LB agar (15 g/L) unless indicated. Mueller Hinton II broth (Oxoid) was used to verify that MIC values of *E. coli* MG1655 were comparable with those obtained in LB medium. The antibiotics used were amoxicillin-clavulanic acid, ceftazidime, cefotaxime, gentamicin (Normon), ertapenem (Merck Sharp and Dohme), kanamycin, chloramphenicol, tetracycline (Nzytech), tigecycline (Pfizer), azithromycin, colistin (Altan Pharmaceuticals), and rifampicin (Sanofi-aventis).

Plasmids pKAZ3 (Accession Number KR827392.1), pKP-M1144 (Acc. No. KF745070.2), pKA2Q (Acc. No. MT720904), and pEncl-30969cz (Acc. No. MG049738.1) were previously published (*Flach et al., 2015*; *Hernández-García et al., 2018a*; *Papagiannitsis et al., 2015*; *Papagiannitsis et al., 2018*). Plasmid pCEMR is a derivative of pEncl-30969cz in which a spontaneous deletion of ~19 kb had occurred (Acc. No. MT720903). Plasmids pOXA-48_K8 (Acc. No. MT441554) and pCF12 (Acc. No. MT720906) were isolated from rectal swabs of patients hospitalized in the Ramón y Cajal University Hospital in the framework of the R-GNOSIS project (*Hernández-García et al., 2018b*). pOXA-48_K8 is a pOXA-48-like plasmid that shares a 100% coverage and >99% identity with the first described pOXA-48 plasmid (*León-Sampedro et al., 2020*; *Poirel et al., 2012*). For simplicity, we refer to this plasmid as pOXA-48 throughout the text. Plasmids pKAZ3, pOXA-48 and pCF12 were conjugated into MG1655 using *E. coli* β3914 as donor strain, which is auxotrophic for diaminopimelic acid (*Le Roux et al., 2007*). Transconjugants were selected in LB plates containing ampicillin (100 mg/L) for plasmids pKAZ3 and pOXA-48 or aztreonam (25 mg/L) for pCF12 plasmid. Plasmids pCEMR, pKP-M1144 and pKA2Q were purified using a commercial mini-prep kit (Macherey-Nagel) and transformed into calcium chloride competent MG1655 cells. Transformant cells were selected on LB-ampicillin plates (100 mg/L). Strain constructions were verified by assessing that the ABR profile of plasmid-bearing MG1655 matched the expected resistance of the plasmid (*Supplementary file 1A*) and by sequencing the complete genomes of transconjugant clones, which also revealed that no significant chromosomal mutations accumulated during the process of plasmid acquisition (0–2 mutations between plasmid-carrying and plasmid-free complete genomes, see *Supplementary file 1C*).

The clinical *E. coli* strains used in this study were isolated during the R-GNOSIS project, which included representative ESBL-producing enterobacteria from hospitalized patients in the Ramon y Cajal University Hospital during a 2-year period (*Hernández-García et al., 2018b*; *León-Sampedro et al., 2020*). The *E. coli* strains used here, and their corresponding transconjugants, were described and characterized previously (*Alonso-del Valle et al., 2020* and *Supplementary file 1D*).

### Antimicrobial susceptibility testing

For the initial screening, single colonies of plasmid-free and plasmid-carrying bacteria were inoculated in LB starter cultures and incubated at 37°C for 16 hr at 225 rpm (4–5 biological replicates performed on different days). Each culture was diluted 1:2000 in LB medium (~5·10$^4$ colony forming units) and 200 µl were added to a 96-well microtiter plate containing the appropriate antibiotic concentration. MIC values were measured after 22 hr of incubation at 37°C. Optical density at 600 nm (OD$_{600}$) was determined in a Synergy HTX (BioTek) plate reader after 30 s of orbital shaking. MIC values corresponded with the lowest antibiotic concentration that resulted in OD$_{600}$ <0.2. We found that this threshold is qualitatively consistent with standard practice (i.e. visual determination of

growth inhibition), while providing higher accuracy and repeatability. MIC determinations of plasmid-free and plasmid-carrying strains were performed in parallel to ensure reproducibility. Fold change in MIC was determined as the ratio between the median MIC from plasmid-carrying derivatives and that from the wild-type strain. To improve the robustness of our results, we performed five to six additional MIC determinations (to a total of n = 10) for the eight plasmid-antibiotic combinations that produced a $\geq 2$ x reduction in ABR in the preliminary screening (*Supplementary file 1A*).

## Disk-diffusion susceptibility testing

LB plates were swabbed with a 0.5 McFarland matched bacterial suspension and sterile disks containing the antibiotic were placed on the plates. Plates were incubated at 37°C for 22 hr and pictures were taken using an in-house photographic system and a cell phone (Huawei Mate 20). Inhibition halos were measured using ImageJ software. Antibiotic disk content used in diffusion assays were as follows (all from Bio-Rad): gentamicin 10 µg, azithromycin 15 µg, kanamycin 30 µg, colistin 10 µg, and tetracycline 30 µg.

## Bacterial growth curves

Starter cultures were prepared and incubated as described above. Each culture was diluted 1:2000 in LB medium and 200 µl were added to a 96-well microtiter plate containing the appropriate antibiotic concentration. Plates were incubated 24 hr at 37°C with strong orbital shaking before reading $OD_{600}$ every 10 min in a Synergy HTX (BioTek) plate reader. Six biological replicates were performed for each growth curve. The area under the growth curve was obtained using the '*auc*' function from the '*flux*' R package. Data was represented using a R custom script and the '*ggplot2*' package.

## Competition assays

Competition assays were performed to measure the relative fitness of plasmid-carrying strains using flow cytometry as previously described (*Rodriguez-Beltran et al., 2018*). Briefly, competitions were performed between MG1655 clones carrying each of the six plasmids and plasmid-free MG1655 against a MG1655 derivative carrying an arabinose inducible chromosomal copy of *gfp* (MG1655::*gfp*). Pre-cultures were incubated at 37°C with 225 rpm shaking overnight in 96-well plates carrying 200 µl of LB broth per well. Pre-cultures were mixed at 1:1 proportion and diluted 1:400 in fresh media. Initial proportions of GFP and non-fluorescent competitors were confirmed in a CytoFLEX Platform (Beckman Coulter Life Sciences) flow cytometer, recording 10,000 events per sample. To measure these proportions, we incubated a culture aliquot in NaCl 0.9% containing 0.5% L-arabinose for 1.5 hr to induce the expression of the chromosomal GFP. Mixtures were competed for 22 hr in LB medium at 37°C with shaking (225 rpm). Final proportions were estimated again by flow cytometry as described above. The fitness of each strain relative to MG1655::*gfp* was calculated using the formula: $w = \ln(N_{final,gfp-}/N_{initial,gfp-})/\ln(N_{final,gfp+}/N_{initial,gfp+})$ where $w$ is the relative fitness of the non GFP-tagged strain, $N_{initial,gfp-}$ and $N_{final,gfp-}$ are the numbers of non GFP-tagged cells before and after the competition and $N_{initial,gfp+}$ and $N_{final,gfp+}$ are the numbers of MG1655::*gfp* cells before and after the competition. To account for the possible cost of *gfp* insertion and/or its expression, plasmid-free MG1655 was competed against MG1655::*gfp* and the data was normalized by dividing the relative fitness of plasmid-carrying strains by the relative fitness obtained for plasmid-free MG1655. Six biological replicates were performed for each competition. We performed control experiments mimicking the growth conditions of the competition experiments to ensure that plasmid loss in plasmid-carrying strains is negligible. Moreover, previous results indicated that plasmid conjugation under the experimental conditions used for the competition assays do not significantly affect relative fitness determinations (*Alonso-del Valle et al., 2020*).

## CS-informed antibiotic treatments

Overnight bacterial cultures were diluted and seeded into 24 independent wells of a 96-well plate filled with 200 µl of LB. After 16 hr of incubation at 37°C, plasmid-free or plasmid-carrying bacterial populations were diluted 1:2000 into fresh medium containing either azithromycin (8 mg/L) or colistin (4 mg/L) and grown separately at 37°C for 22 hr. Although we used non-standardized media and culture conditions, these concentrations are within the range of clinical breakpoints determined by

EUCAST, which are 16 µg/ml for azithromycin (clinical breakpoints only available for *Salmonella typhi* and *Shigella* spp) and 2 µg/ml for colistin (according to EUCAST clinical breakpoints version eleven; *EUCAST, 2021*). The following day, bacterial populations were diluted (1:2000) and inoculated into new plates containing, again, either azithromycin or colistin and allowed to grow as above. This approach led to four antibiotic treatments, two in which the antibiotic remains constant (Azi→ Azi and Col→ Col) and two in which the antibiotic treatment alternates (Azi→ Col and Col→ Azi). The growth of bacterial populations was assessed by measuring $OD_{600}$ every day. Populations that did not reach an $OD_{600}$ of 0.2 were declared extinct.

## Whole genome sequencing

Genomic DNA was isolated using the Wizard genomic DNA purification kit (Promega) and quantified using the QuantiFluor dsDNA system (Promega) following manufacturer's instructions. Whole genome sequencing was conducted at the Wellcome Trust Centre for Human Genetics (Oxford, UK), using the Illumina HiSeq4000 platform with 125 base pair (bp) paired-end reads. For plasmid pCF12, first described in this study, we performed additional long-read sequencing using PacBio technologies. PacBio sequencing was performed at The Norwegian Sequencing Centre PacBio RSII platform using P5-C3 chemistry. Illumina and PacBio sequence reads were trimmed using the Trimmomatic v0.33 tool (*Bolger et al., 2014*). SPAdes v3.9.0 was used to generate *de novo* assemblies from the trimmed sequence reads with the –cov-cutoff flag set to 'auto' (Bankevich et al., 2012). Unicycler was used to generate hybrid assemblies from Illumina and PacBio data (*Wick et al., 2017*). QUAST v4.6.0 was used to generate assembly statistics (*Gurevich et al., 2013*). All the *de novo* assemblies reached enough quality including total size of 5–7 Mb, and the total number of contigs over 1 kb was lower than 200. Prokka was used to annotate the *de novo* assemblies (*Seemann, 2014*). The plasmid content of each genome was characterised using PlasmidFinder 2.1 (*Carattoli et al., 2014*), and the ABR gene content was characterised with ResFinder 3.2 (*Zankari et al., 2012*). MOB families were characterised using MOB-typer tool included in MOB-suite (*Robertson and Nash, 2018*). Variant calling was performed using Snippy v4.6.0 (https://github.com/tseemann/snippy). The sequences generated and analysed during the current study are available in the Sequence Read Archive (SRA) repository, BioProject ID: PRJNA644278 (https://www.ncbi.nlm.nih.gov/bioproject/644278), and Genbank accession numbers are provided in *Table 1*.

To determine the distribution of the *E. coli* isolates across the phylogeny of the species, we obtained 1334 assemblies of *E. coli* complete genomes from the RefSeq database (https://www.ncbi.nlm.nih.gov/assembly). Distances between genomes were established using Mash v2.0 (*Ondov et al., 2016*) and a phylogeny was constructed with mashtree v0.33 (*Katz et al., 2019*). The tree was represented with midpoint root using the phytools package in R and visualised using the iTOL tool (*Letunic and Bork, 2011*). To determine the phylotype of each genome, we used the ClermonTyping tool (*Beghain et al., 2018*).

## Statistical analyses

Data sets were analyzed using Prism six software (GraphPad Software Inc) and R. Normality of the data was assessed by visual inspection and the Shapiro-Wilk test. Mann-Whitney U test was used to assess the significance of MIC determinations (n = 10) for the 7 CS instances obtained after the preliminary screening. ANOVA test were perfomed to ascertain the effect of the 'plasmid x antibiotic concentration' interaction term in the analysis of growth curves (area under the growth curve). Statistical analyses of the disk-diffusion halos were performed using unpaired t-tests with Welch's correction. Survival curves were analyzed using the log-rank test within the '*survminer*' R package.

## Acknowledgements

We are grateful to thank Costas Papagiannitsis, Carl-Frederik Flach, Fabrice Elia Graf and Martin Palm for generous gifts of plasmids. We also appreciate the technical support of Carmen de la Vega and Laura Jaraba. This work was supported by the European Research Council under the European Union's Horizon 2020 research and innovation programme (ERC grant agreement no. 757440-PLAS-REVOLUTION) and by the *Instituto de Salud Carlos III* (grant PI16-00860) co-funded by European Development Regional Fund 'a way to achieve Europe'. RC acknowledges financial support from European Commission (grant R-GNOSIS-FP7-HEALTH-F3-2011-282512) and *Plan Nacional de I+D*

+i2013–2016 and *Instituto de Salud Carlos III, Subdirección General de Redes y Centros de Investigación Cooperativa, Ministerio de Economía, Industria y Competitividad*, Spanish Network for Research in Infectious Diseases (REIPIR D16/0016/0011) co-financed by European Development Regional Fund 'A way to achieve Europe' (ERDF), Operative program Intelligent Growth 2014–2020. ASM is supported by a Miguel Servet Fellowship (MS15-00012). JRB is a recipient of a *Juan de la Cierva-Incorporación* Fellowship (IJC2018-035146-I) co-funded by *Agencia Estatal de Investigación del Ministerio de Ciencia e Innovación*. CH is supported by *Comunidad Autónoma de Madrid* (PEJD-2018-POST/BMD-8016).

## Additional information

### Funding

| Funder | Grant reference number | Author |
| --- | --- | --- |
| European Research Council | ERC-StG 757440-PLASREVOLUTION | Álvaro San Millán |
| Instituto de Salud Carlos III | PI16-00860 | Álvaro San Millán |
| Agencia Estatal de Investigación | IJC2018-035146-I | Jerónimo Rodríguez-Beltrán |
| Instituto de Salud Carlos III | MS15-00012 | Álvaro San Millán |
| Comunidad Autónoma de Madrid | PEJD-2018-POST/BMD-8016 | Cristina Herencias |
| European Commission | R-GNOSIS-FP7-HEALTH-F3-2011-282512 | Rafael Cantón |
| Instituto de Salud Carlos III | REIPIR D16/0016/0011 | Rafael Cantón |

The funders had no role in study design, data collection and interpretation, or the decision to submit the work for publication.

### Author contributions

Cristina Herencias, Data curation, Formal analysis, Investigation, Methodology, Writing - original draft, Writing - review and editing; Jerónimo Rodríguez-Beltrán, Conceptualization, Formal analysis, Investigation, Methodology, Writing - original draft, Writing - review and editing; Ricardo León-Sampedro, Data curation, Formal analysis, Investigation, Writing - review and editing; Aida Alonso-del Valle, Resources, Writing - review and editing; Jana Palkovičová, Investigation, Writing - review and editing; Rafael Cantón, Resources, Investigation, Writing - review and editing; Álvaro San Millán, Conceptualization, Supervision, Funding acquisition, Investigation, Methodology, Writing - original draft, Project administration, Writing - review and editing

### Author ORCIDs

Cristina Herencias https://orcid.org/0000-0002-1384-3109
Jerónimo Rodríguez-Beltrán https://orcid.org/0000-0003-3014-1229
Ricardo León-Sampedro https://orcid.org/0000-0001-5317-8310
Aida Alonso-del Valle https://orcid.org/0000-0002-5154-5778
Jana Palkovičová https://orcid.org/0000-0003-4836-5239
Rafael Cantón https://orcid.org/0000-0003-1675-3173
Álvaro San Millán https://orcid.org/0000-0001-8544-0387

### Decision letter and Author response

Decision letter https://doi.org/10.7554/eLife.65130.sa1
Author response https://doi.org/10.7554/eLife.65130.sa2

## Additional files

### Supplementary files

• Supplementary file 1. This file includes antibiotic susceptibility data. (A) ANOVA results of the growth curves (B), information on chromosomal mutations accumulated during plasmid acquisition (C), and the characteristics of the *E. coli* clinical isolates used in this study (D).

• Transparent reporting form

### Data availability

All data generated or analysed during this study are included in the manuscript and supporting files. Source data files have been provided for Figures 1, 2, 3 and S2. Sequencing data have been deposited in the Sequence Read Archive (SRA) repository, BioProject ID: PRJNA644278 (https://www.ncbi.nlm.nih.gov/bioproject/644278).

The following dataset was generated:

| Author(s) | Year | Dataset title | Dataset URL | Database and Identifier |
|---|---|---|---|---|
| Herencias C, Rodríguez-Beltrán J, León-Sampedro R, Valle AA-D, Palkovičová J, Cantón R, Millán SA | 2020 | DNA Sequence Data | https://www.ncbi.nlm.nih.gov/bioproject/PRJNA644278 | NCBI BioProject, PRJNA644278 |

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
