## [Decision Letter]

**Acceptance summary:**

This is an important contribution to documenting the relevance of collateral sensitivity in plasmid-borne resistance.

**Decision letter after peer review:**

Thank you for submitting your article "Collateral sensitivity associated with antibiotic resistance plasmids" for consideration by *eLife*. Your article has been reviewed by three peer reviewers, one of whom is a member of our Board of Reviewing Editors, and the evaluation has been overseen by Dominique Soldati-Favre as the Senior Editor. The following individual involved in review of your submission has agreed to reveal their identity: Adam C Palmer (Reviewer #3).

The reviewers have discussed the reviews with one another and the Reviewing Editor has drafted this decision to help you prepare a revised submission.

Summary:

This paper describes very clearly a set of experiments to assess collateral sensitivity to certain antibiotics that is created by carriage of β-lactam (including carbapenenem) resistance plasmids in *E. coli*. This addresses one of the limitations of existing literature on CS, which typically focuses on the effects of resistance point mutations, which are clinically less significant. By documenting multiple ways that this CS is real and selectable and to a degree generalizable across genetic backgrounds, this is an important contribution in showing that CS is a real phenomenon for clinically important resistance mechanisms.

Essential revisions:

1) The primary screen of “antibiotics x plasmids” to identify collateral sensitivity, presented in Figure 1B, lacks an analysis of the statistical significance of results. Supplementary data shows that measurements of MIC are a little too noisy to robustly identify 2-fold changes with only 4 or 5 replicates. Defining "significant" as "mean more than 2x" is not adequate. Using a power calculation derived from the data in the manuscript, a sample size should be determined to have a 90 (or other high) % chance of detecting a 2x difference given the variability observed between assays, and then they should be done. Ideally this would be for all organism-plasmid pairs, but at least for the ones that the preliminary screen found a mean of 2x for.

2) Recommendation (not required for acceptance, but please temper claims of clinical relevance if not done): The comparative killing data should be repeated in competition. This is technically more challenging but I believe not more so than the comparative growth curves. This would establish as proof of principle that a mixed population could be purified of plasmid-bearers by CS. Without this, the clinical relevance will still remain speculative. Also, two reviewers initially misread these as competition assays. The text and legend should emphasize that these are separate populations

3) (not required for acceptance but suggestion for future work): The presented work is solid but, as pointed out by the authors, there is no mechanistic explanations for the observations. It would be highly interesting to know if the collateral effects are due to specific genes (OXA-48 would be a good place to start) and/or if the observed effects are due to the plasmid backbone.

4) The experiment in Figure 3 demonstrates the exploitation of collateral sensitivity to preferentially inhibit plasmid-bearing bacteria. The terminology in this section refers to “eradicate”, “mortality” etc, but in practice, the experiment defines survival as OD>0.2 after ~24 hours. It seems likely that in the “non-surviving” conditions, waiting another day or two would show regrowth of some bacteria in these conditions. We don't think this requires any change to the experiment, only how the results are described: they show preferential inhibition of growth, not eradication. A more patient approach to identifying regrowth would be necessary to definitively state that these bacterial populations have been eradicated. Suggest tempering claims.

[Editors' note: further revisions were suggested prior to acceptance, as described below.]

Thank you for resubmitting your work entitled "Collateral sensitivity associated with antibiotic resistance plasmids" for further consideration by *eLife*. Your revised article has been evaluated by Dominique Soldati-Favre (Senior Editor) and a Reviewing Editor.

The manuscript has been improved but there are some remaining issues that need to be addressed before acceptance, as outlined below:

The revision is excellent. In my opinion you might perhaps have gone slightly too far in accommodating reviewer comments, removing all references to killing and replacing them with growth inhibition. I think it is clear that considerable selective killing happens; the reviewers objected to the notion of eradication. I fear that the use of "growth inhibition" will confuse the reader into thinking the effect is purely bacteriostatic (and perhaps partial). Can you please use "killing" where appropriate and avoid "eradication" or similar words? In summary, the individual cells are killed, not just inhibited, but the culture may not be cleared (100% killing, eradication).

---

## [Author Response]

Essential revisions:1) The primary screen of “antibiotics x plasmids” to identify collateral sensitivity, presented in Figure 1B, lacks an analysis of the statistical significance of results. Supplementary data shows that measurements of MIC are a little too noisy to robustly identify 2-fold changes with only 4 or 5 replicates. Defining "significant" as "mean more than 2x" is not adequate. Using a power calculation derived from the data in the manuscript, a sample size should be determined to have a 90 (or other high) % chance of detecting a 2x difference given the variability observed between assays, and then they should be done. Ideally this would be for all organism-plasmid pairs, but at least for the ones that the preliminary screen found a mean of 2x for.

We agree with the reviewers that in the previous version of the manuscript, MIC results were not statistically evaluated. As suggested, we performed power analyses for the combinations where we detected a difference of at least 2x in the initial analysis, assuming normality of the data [as nonparametric power analysis is not trivial (Shieh et al., 2006)]. The results show that 10-50 replicates would be required to statistically validate this difference (at α=0.05 and a power of 0.80; https://www.stat.ubc.ca/~rollin/stats/ssize/n2.html). We have now performed additional MIC determinations (total of 10 biological replicates) and statistical tests for the plasmid-antibiotic combinations that were identified in the preliminary screening. New data using 10 biological replicates shows that 6 out of the 8 originally described CS instances are indeed significant (MannWhitney U test; P<0.015 in all cases). For the remaining 2 non-significant CS instances, one (pCF12-tetracycline) does not reach the ≥2x threshold after combining data from all replicates, so we decided to remove it from subsequent analyses. The other non-significant event (pKA2Qtetracycline) is above the 2x threshold, but the difference is not statistically significant (MannWhitney U test; P=0.2039). The results of the power analysis showed that 50 replicates per genotype (MG/pKA2Q and MG1655) would be required to statistically validate this difference. Given that the CS produced by pKA2Q is statistically validated with both growth curves and disk assays, we honestly believe that performing that many replicates is unnecessary.

These changes have been introduced in the text and in the figure legend where we now state that MIC determinations were first performed as a preliminary screening and we report the statistical significance of the ≥2x CS instances. Additionally, we have modified Figure 1 and Figure 1—figure supplement 1 to reflect the new data.

2) Recommendation (not required for acceptance, but please temper claims of clinical relevance if not done): The comparative killing data should be repeated in competition. This is technically more challenging but I believe not more so than the comparative growth curves. This would establish as proof of principle that a mixed population could be purified of plasmid-bearers by CS. Without this, the clinical relevance will still remain speculative. Also, two reviewers initially misread these as competition assays. The text and legend should emphasize that these are separate populations

We agree that a direct competition experiment would establish a nice proof of principle of the clinical potential of CS to purify plasmid-bearers. In fact, that was the original experimental design for this section, and we even fluorescently tagged the different wild-type strains in order to use flow cytometry to perform high-throughput competitions. However, when we started to run the experiments, we realized that the presence of azithromycin (protein synthesis inhibitor) affects GFP production leading to unreliable results. In addition, when we tried an alternative approach to flow cytometry for the competitions (plating on selective agar), we had the problem of selecting for pOXA-48 plasmid, which gives rise to a strong inoculum effect in the plasmid-bearing clones. As a consequence, plating results became noisy, especially when using the ESBL-producing *E. coli* clinical isolates, which are already resistant to most *β*-lactam antibiotics (the low-level resistance conferred by pOXA-48 to carbapenems exacerbates the inoculum effect issue when selecting in these backgrounds). Finally, we also thought about the possibility of performing the same killing curves we performed, but using mixed populations (plasmid-bearing/plasmid-free) instead of pure cultures. However, since these cultures are propagated over a few days, pOXA-48 conjugation becomes a potential problem for the interpretation of the results. Therefore, we finally decided to use the alternative approach of comparative killing curves with pure cultures. We believe that these experiments, although far from being perfect, provide a nice proof of concept of the potential clinical exploitation of CS effects associated with AR plasmid acquisition. However, we agree with the reviewers that the clinical relevance of our study still remain speculative, and that in the previous version of the manuscript we overstated the clinical implications of our results, so we have toned down those claims through the manuscript and included a sentence to clearly state the limitations of our experimental design in the concluding remarks section. We have also improved the description of the experiment in the figure legend, and the Results and Materials and methods sections to better convey the idea that plasmid-bearers and plasmid-free populations were subjected to antibiotic treatments separately.

3) (not required for acceptance but suggestion for future work): The presented work is solid but, as pointed out by the authors, there is no mechanistic explanations for the observations. It would be highly interesting to know if the collateral effects are due to specific genes (OXA-48 would be a good place to start) and/or if the observed effects are due to the plasmid backbone.

Among the several lines of research that our work opens, understanding the molecular causes of CS is arguably one of the most interesting. Indeed, we are undertaking a new project in which we take advantage of our extensive collection of pOXA-48-carrying clinical isolates (León-Sampedro et al., 2020) to gain insight into these mechanistic details. In particular, we are testing if different pOXA-48 variants carrying deletions that affect different plasmid traits (conjugation, toxin-antitoxin, blaOXA-48…) reduce or abolish the CS phenotype. Additionally, we are performing transcriptomic analyses of clinical strains to gain insight into the metabolic perturbations that the plasmid produces in the host cell. We are confident that these experiments will suggest why pOXA-48 produces the observed CS effects, but extensive work will still be needed to generalize these results to other plasmids.

4) The experiment in Figure 3 demonstrates the exploitation of collateral sensitivity to preferentially inhibit plasmid-bearing bacteria. The terminology in this section refers to “eradicate”, “mortality” etc, but in practice, the experiment defines survival as OD>0.2 after ~24 hours. It seems likely that in the “non-surviving” conditions, waiting another day or two would show regrowth of some bacteria in these conditions. We don't think this requires any change to the experiment, only how the results are described: they show preferential inhibition of growth, not eradication. A more patient approach to identifying regrowth would be necessary to definitively state that these bacterial populations have been eradicated. Suggest tempering claims.

We completely agree with the reviewers that the experiment shows growth inhibition rather than eradication and that regrowth might occur under the appropriate circumstances. We have removed any reference to mortality or eradication of plasmid-carrying bacteria thorough the paper, and we now interpret this results in terms of growth inhibition, as suggested.

[Editors' note: further revisions were suggested prior to acceptance, as described below.]

The manuscript has been improved but there are some remaining issues that need to be addressed before acceptance, as outlined below:The revision is excellent. In my opinion you might perhaps have gone slightly too far in accommodating reviewer comments, removing all references to killing and replacing them with growth inhibition. I think it is clear that considerable selective killing happens; the reviewers objected to the notion of eradication. I fear that the use of "growth inhibition" will confuse the reader into thinking the effect is purely bacteriostatic (and perhaps partial). Can you please use "killing" where appropriate and avoid "eradication" or similar words? In summary, the individual cells are killed, not just inhibited, but the culture may not be cleared (100% killing, eradication).

As suggested, in the new version of the manuscript we have replaced all references to growth inhibition with terms related to killing, survival and mortality, which in our opinion offer a more accurate description of the experimental results. We have changed Figure 3 y-axis title accordingly and ensured that we do not refer to our experiment using terms such as “eradication”.